# Gold Glyconanoparticles Combined with 91–99 Peptide of the Bacterial Toxin, Listeriolysin O, Are Efficient Immunotherapies in Experimental Bladder Tumors

**DOI:** 10.3390/cancers14102413

**Published:** 2022-05-13

**Authors:** Hector Terán-Navarro, Andrea Zeoli, David Salines-Cuevas, Marco Marradi, Noemi Montoya, Elena Gonzalez-Lopez, Javier Gonzalo Ocejo-Vinyals, Mario Dominguez-Esteban, Jose Luis Gutierrez-Baños, Felix Campos-Juanatey, Sonsoles Yañez-Diaz, Almudena Garcia-Castaño, Fernando Rivera, Ignacio Duran, Carmen Alvarez-Dominguez

**Affiliations:** 1Grupo de Oncología y Nanovacunas, Instituto de Investigación Marqués de Valdecilla (IDIVAL), 39011 Santander, Cantabria, Spain; hteran35@hotmail.com (H.T.-N.); andrea.zeoli@libero.it (A.Z.); davidsalcines@gmail.com (D.S.-C.); sonsolesjuana.yanez@scsalud.es (S.Y.-D.); 2Dipartimento di Chimica “Ugo Schiff”, Universitá Degli Studi di Firenze, I-50019 Firenze, Italy; marco.marradi@unifi.it; 3Grupo de Investigación MEDONLINE, Facultad de Ciencias de la Salud, Universidad Internacional de La Rioja, Avda. de La Paz 137, 26006 Logroño, La Rioja, Spain; noemi.montoya@unir.net; 4Servicio de Inmunología, Hospital Universitario Marqués de Valdecilla, Instituto de Investigación Marqués de Valdecilla (IDIVAL), 39008 Santander, Cantabria, Spain; elena.gonzalez@scsalud.es (E.G.-L.); javiergonzalo.ocejo@scsalud.es (J.G.O.-V.); 5Servicio de Urología, Hospital Universitario Marqués de Valdecilla, Instituto de Investigación Marqués de Valdecilla (IDIVAL), 39008 Santander, Cantabria, Spain; mario.dominguez@scsalud.es (M.D.-E.); joseluis.gutierrez@scsalud.es (J.L.G.-B.); felix.campos@scsalud.es (F.C.-J.); 6Servicio de Dermatología, Hospital Universitario Marqués de Valdecilla, Instituto de Investigación Marqués de Valdecilla (IDIVAL), 39008 Santander, Cantabria, Spain; 7Servicio de Oncología Médica, Hospital Universitario Marqués de Valdecilla, Instituto de Investigación Marqués de Valdecilla (IDIVAL), 39008 Santander, Cantabria, Spain; almudena.garcia@scsalud.es (A.G.-C.); fernando.rivera@scsalud.es (F.R.); ignaciojose.duran@scsalud.es (I.D.)

**Keywords:** listeriolysin O, melanoma, bladder cancer, nanoparticles, immunotherapy

## Abstract

**Simple Summary:**

We propose a novel type of immunotherapy for bladder cancer using gold nanoparticles bound to a peptide of a bacterial toxin with anti-tumor capacities as listeriolysin O called Listeria nanovaccines. Here, we present the pre-clinical experiments on a mice model of bladder cancer and blood cells of patients with bladder cancer using these Listeria nanovaccines that activate the immune system, block the tumor immunosuppression environment, and reduce the tumor size. The impact of Listeria nanovaccines on the field of immunotherapies for solid tumors can be extended to other solid tumors containing lymphocyte infiltration. Therefore, we propose Listeria nanovaccines as immunotherapy for tumors such as melanoma, urothelial bladder carcinoma, non-small cell lung carcinoma, and glioblastoma.

**Abstract:**

This study presents proof of concept assays to validate gold nanoparticles loaded with the bacterial peptide 91–99 of the listeriolysin O toxin (GNP-LLO_91–99_ nanovaccines) as immunotherapy for bladder tumors. GNP-LLO_91–99_ nanovaccines showed adjuvant abilities as they induce maturation and activation of monocyte-derived dendritic cells (MoDCs) to functional antigen-presenting cells in healthy donors and patients with melanoma or bladder cancer (BC), promoting a Th1 cytokine pattern. GNP-LLO_91–99_ nanovaccines were also efficient dendritic cell inducers of immunogenic tumor death using different bladder and melanoma tumor cell lines. The establishment of a pre-clinical mice model of subcutaneous BC confirmed that a single dose of GNP-LLO_91–99_ nanovaccines reduced tumor burden 4.7-fold and stimulated systemic Th1-type immune responses. Proof of concept assays validated GNP-LLO_91–99_ nanovaccines as immunotherapy by comparison to anti-CTLA-4 or anti-PD-1 antibodies. In fact, GNP-LLO_91–99_ nanovaccines increased percentages of CD4^+^ and CD8^+^ T cells, B cells, and functional antigen-presenting DCs in tumor-infiltrated lymphocytes, while they reduced the levels of myeloid-derived suppressor cells (MDSC) and suppressor T cells (T_reg_). We conclude that GNP-LLO_91–99_ nanovaccines can work as monotherapies or combinatory immunotherapies with anti-CTLA-4 or anti-PD-1 antibodies for solid tumors with high T cell infiltration, such as bladder cancer or melanoma.

## 1. Introduction

In the past 10 years, nanotechnology has been widely studied for cancer treatment, as nanoparticles can play a significant role in drug delivery systems [1]. In this regard, inorganic nanoparticles such as gold nanoparticles (AuNP) are useful candidates since the gold core is inert and non-toxic and can be surface-functionalized with different compounds. Functionalization with carbohydrates might enhance their accumulation into tumors and overcome drug resistance. In addition, the AuNP combination with peptides might target them to specific cells or confer novel features [2,3]. In this regard, gold glyconanoparticles (AuGNP)—here, GNP for simplicity—present several advances such as high water solubility, ability to target antigen-presenting cells (APC), lack of toxicity [4,5], and capacity to combine in the same design with different ligands, such as carbohydrates and peptides [6,7].

Adjuvants are classical immunotherapies that activate the tumoral immune responses by acting on antigen-presenting cells (APCs) (e.g., dendritic cells, DCs); examples are inorganic nanoparticles, cancer vaccines, oligonucleotides, bacterial compounds, and antagonists of Toll-like receptors [8,9]. However, the development of immune checkpoint inhibitors (ICIs) (e.g., anti-CTLA-4 or anti-PD-1/PD-L1 antibodies) as new immunotherapies changed the focus of immune system activation to T cells, as they block the negative immune controls of T cells and boost the anti-tumoral immune responses [10]. ICIs have been approved as either a neoadjuvant or first-line treatment for tumors such as melanoma, non-small cell lung (NSCLC), renal, or bladder cancer (BC). In general, ICIs seem to benefit oncologic patients with tumors presenting high T cell infiltration and mutational variance. However, in some cases, ICI resistance or adverse effects minimize their efficiency [11], along with the failure to respond to ICI of tumors with low-medium mutational variance (e.g., hepatocarcinoma) or infiltrated T cells (e.g., glioblastoma) [12,13,14,15,16].

Bladder cancer (BC) is the ninth most common malignancy diagnosed worldwide. As a tumor, BC is highly immunogenic and T cell-infiltrated. It has been postulated that a dysregulation of the immune system within the bladder promotes BC growth. However, to date, this immunosuppression within BC has not been characterized in detail [17]. BC arises from the urothelium, the epithelium that lines the urinary bladder, and only 25% at most are muscle-invasive metastatic BCs. Metastatic BC is treated with chemotherapy regimens as first-line treatments and, recently, ICI as neoadjuvant, maintenance or second-line treatment [18]. Most BCs are superficial, low-grade, noninvasive, or superficial tumors confined to the mucosa layer. Noninvasive BC standard therapy implies tumor resection followed by adjuvant therapy with the bacterium Bacillus Calmette-Guerin (BCG) to activate the immune system [19,20]. In this regard, *Listeria monocytogenes* is a human bacterial pathogen also used for the treatment of several tumors such as prostate cancer, cervix carcinoma, or pancreatic cancer, using attenuated mutants that lack the C-terminal domain of the main virulence factor, listeriolysin O (LLO) [21,22,23]. Moreover, LLO is a pore-forming toxin that, along with its cytolytic activity, induces cell death of different cell types such as DCs, macrophages, or T cells [24]. LLO seems to induce different types of cell death, necrosis, necroptosis, pyroptosis, or apoptosis, causing different effects [25]. The classical assumption that necrosis induces inflammatory immune responses while apoptosis triggers anti-inflammatory immune responses does not seem applicable in all contexts of cell death by bacterial toxins. While the tuberculosis-necrotizing toxin (TNT) induces immunological silent cell death that recruits macrophages and induces poor immune responses [26], LLO action onto tumor cells appears to induce an immunogenic cell death characterized by promoting inflammatory and anti-tumor immune responses. Previous work of our group quoted LLO anti-neoplastic abilities to 91–99 N-terminal peptides and explored different vaccine vectors as therapy for experimental melanoma [5,6,7]. Here, we extend these studies using a nanocarrier vector GNP coupled to two ligands, the 91–99 LLO peptide and β-D-glucose (here called GNP-LLO_91–99_ nanovaccines), to validate them as valid immunotherapies for bladder tumors and inducers of immunogenic apoptosis.

## 2. Materials and Methods

### 2.1. Cells and Mice

B16.F10 murine metastatic melanoma, A375 human melanoma, MB-49 murine and human T-24 bladder tumor cell lines, TC-1 murine and A-549 NSCLC lung tumor cell lines, O627 murine glioblastoma cell lines, Hepa1-6 murine hepatocarcinoma cell lines, CHO hamster ovary cell lines and SV-40-induced IC-21 murine macrophage cell lines were obtained from ATCC (Manassas, VA, USA). The RG-1 human glioblastoma cell line was a gift of J. L. Fernandez-Luna (HUMV, Santander, Spain). C57BL/6 male or female mice (Charles River, L’Abresle, France) 8–12 months of age were used in the study.

### 2.2. Patients

Nine patients with advanced solid tumors were included in this study: three patients with stage IV melanoma before enrollment in immunotherapy (MEL-1 and MEL-3), a stage IIIB cutaneous melanoma after surgery resection (MEL-2), a lung and bladder carcinoma treated with cisplatin-etoposide (BC-1), two urothelial bladder carcinomas without treatment (BC-2, BC-3), a hepatocellular carcinoma treated with ablation by microwaves (HEP-1), a prostate adenocarcinoma treated with taxocel (PROST-1), and a multiform glioblastoma treated with temozolomide and radiotherapy (GLIO-1). MEL-1, MEL-3, BC-1, HEP-1, PROST-1, and GLIO-1 were diagnosed at the Medical Oncology Department; BC-2 and BC-3 were diagnosed at the Urology Department; and MEL-2, the stage IIIB cutaneous melanoma, was diagnosed at the Dermatology Department (Hospital U. Marqués de Valdecilla, HUMV, Santander, Spain). The patients participated in the study voluntarily, signed an informed consent agreement at the physician consultation, and received an information document on the research study. Patients were able to revoke the informed consent at any time. Healthy donors were obtained from the blood bank at our hospital. Blood samples were obtained in heparin tubes at the Dermatology, Urology, or Medical Oncology Departments on the day of patient consultation and processed at the IDIVAL laboratory within the following 2 h.

### 2.3. Preparation of GNP-LLO_91–99_ Nanovaccines

LLO_91–99_ peptide with a C-terminal cysteamide (LLO_91–99_C(O)NHCH_2_CH_2_SH, 14 mg, purity 95%) was purchased from GenScript (Piscataway, NJ, USA) and 5-(mercapto)pentyl-β-D-glucopyranoside (GlcC_5_SH) was prepared as reported previously by our group [27]. GNPs carrying 90% glucose and 10% LLO_91–99_ peptide were prepared by the reduction of Au(III) salt with sodium borohydride following previously described procedures [28]. Peptide loading was GNP-LLO_91–99_: 8.9 µg (LLO_91–99_)/0.182 mg of GNPs. In brief, LLO_91–99_C(O)NHCH_2_CH_2_SH (1 mg, 0.85 µmol) and GlcC_5_SH (2.1 mg, 7.4 µmol) were dissolved in deuterated water (750 µL). ^1^H NMR analysis of the mixture showed a Glc:LLO_91–99_ ratio of ~9:1 (e.g., 90% Glc and 10% LLO_91–99_). The above-prepared solution of the ligands (0.011 M, 4 equivalents) was added with an aqueous solution of HAuCl_4_ (100 μL, 0.025 M, 1 equivalent), followed by an aqueous NaBH_4_ solution (67.5 μL, 1 M, 27 equivalents) in four portions under rapid shaking. The dark dispersion was shaken for 2 h and then filtered with 3 KDa MWCO tubes by centrifugal filtering. The black colloid was recovered with water and lyophilized (0.482 mg). The residue was re-dispersed in the minimum volume of water, loaded in SnakeSkin pleated dialysis tubing (Pierce, 10 KDa MWCO), and dialyzed against 3 L of water under gentle stirring, recharging with fresh distilled water every ~8 h over the course of 72 h. After lyophilization, 0.456 mg of GNP-LLO_91–99_ were obtained. The ratio of glucose/peptide in the GNP was determined by quantitative NMR (qNMR) in a Bruker AVANCE 500 MHz spectrometer: 0.456 mg of GNP-LLO_91–99_ was dispersed in D_2_O 99.9% (200 µL). Then, 80 µL of this solution was added to 40 µL of a 0.05% 3-(trimethylsilyl)propionic-2,2,3,3-d_4_ acid sodium salt (TSP-d4) solution in D_2_O and 60 µL of D_2_O. ^1^H NMR analysis of the mixture allowed the calculation of the amount of peptide in the GNP-LLO_91–99_: 8.9 µg (LLO_91–99_)/0.182 mg (GNP). TEM (JEOL JEM-2100F working at 200 kV). A single drop (5 μL) of the aqueous dispersion (ca. 0.1 mg mL^−1^ in MilliQ water) of the GNPs was placed onto a copper grid coated with an ultrathin carbon film (Electron Microscopy Sciences). The grid was air-dried for 12 h at RT. Average gold diameter was 1.5 ± 0.5 nm, obtained by counting 600 particles in the TEM image as previously reported [28]. UV/Vis (Beckman Coulter DU 800 Spectrometer, H_2_O, 0.1 mg/mL): The average gold core size was confirmed by UV-Vis spectra, which did not show an absorption maximum at around 520 nm, typical of gold nanoparticles with a larger core diameter [29].

### 2.4. Isolation of MoDCs from Healthy Donors and Patients and Preparation of Murine DCs

Leukocytes from whole blood cells were collected from the interphase of a Ficoll gradient and incubated with microbeads conjugated to a mouse IgG2a monoclonal anti-CD14 antibody and passed through MACS^TM^ columns (Miltenyi, Bergisch Gladbach, Germany) to select monocytes (Mo) as CD14^+^ positive cells. Mo (1 × 10^6^ of cells/mL) were cultured in 6-well plates (Falcon^TM^, Corning Life Sciences, Glendale, AZ, USA) over 7 days using GM-CSF (50 ng/mL) and IL-4 (20 ng/mL) in RPMI-20% FCS medium. Differentiated MoDCs were 98% CD45^+^CD11c^+^HLA-DR^+/−^CD86^−^CD14^−^ positive cells. Bone-marrow-derived dendritic cells (here called DC) were obtained from C57BL/6 control or bladder-transplanted mice femurs and differentiated with GM-CSF (20 ng/mL) for 7 days. Differentiated DCs presented a phenotype of 98% CD11c^+^MHC-I^+^MHC-II^+^CD11b^−/+^CD40^−^CD86^−^ cells.

### 2.5. Direct and Immunogenic Apoptosis of Melanoma and Bladder Tumor Cells

Cell toxicity was first evaluated using Trypan blue staining and incubation for 16 h with GNP-LLO_91–99_ nanovaccines. Direct apoptosis was evaluated in the different tumor cell lines after incubation for 16 h with GNP-LLO_91–99_ (50 µg/mL). Immunogenic apoptosis implies the incubation of tumor cell lines with ½ supernatants of ex vivo differentiated DCs (for murine tumors) or MoDCs from healthy donors (for human tumors), pre-treated for 16 h with 50 µg/mL of GNP-LLO_91–99_ nanovaccines. Direct and immunogenic apoptosis was examined by FACS after double staining with the DNA marker, 7-AAD (7-AAD-PE), and the apoptotic marker, annexin V (annexin V-APC). The staining of cells with 7-ADD corresponded to necrotic cell death, whereas the staining of cells with annexin-V alone corresponded to early apoptotic programmed cell death (mean ± SD). Experiments involving human samples were performed three times and five times for mice assays. Results are expressed as the % of apoptotic cells ± SD of triplicate samples (*p* ≤ 0.5).

### 2.6. Bladder Tumor Auto Transplants Followed by GNP-LLO_91–99_ and ICI Immunotherapies

Murine MB-49 bladder cell lines [27] or B16.F10 melanoma were transplanted into 8–12-week-old C57BL/6 male or female mice (auto transplants), respectively, with a single subcutaneous (s.c.) injection (10^6^ cells) in a volume of 100 µL (n = 10/group). At 14 days, tumor transplanted mice received or not (NT) a single intravenous (i.v.) inoculation of GNP-LLO_91–99_ nanovaccines (50 µg/mouse). On day 7, mice were sacrificed, and tumor sizes were measured with a caliper. Values shown for tumor volume (TV) were calculated using the following formula: (length × (width)^2^)/2, as reported [6]. Mean and SD of the tumor volumes per group were calculated. Sera were also obtained, processed, and used for cytokine measurements. Tumors were minced, homogenized, passed through a 70 µm strainer, and then centrifuged over a Ficoll gradient at a 1.077 g/mL density (Histopaque-1077, Sigma-Aldrich, St. Louis, MO, USA) to recover TILs in the interphase gradient while collecting tumor cells in pellets. For immunotherapeutic studies, treatment of MB-49 established transplants with GNP-LLO_91–99_ nanovaccines were performed as above, alone or in combination with the following antibodies: anti-CTLA-4 or anti-PD-1 (50 µg/mouse). Cell populations of TILs and spleens were analyzed by FACS. All experiments were performed at least five times.

### 2.7. FACS Analysis and Cytokines

Cell surface markers of human MoDCs were analyzed by FACS using the following antibodies for human MoDC: anti-HLA-DR-FITC, anti-CD45-PerCP, anti-CD11c-APC, anti-CD14-PE, and anti-CD86-V450 (clone 2331). Murine spleens and TILs were analyzed by FACS using the following antibodies: biotin anti-IAb (clone AF6-120-1), anti-CD11c-PE (clone HL3), anti-CD40-APC (monoclonal 3/23 from BD Pharmingen, BD Biosciences, San Jose, CA, USA), anti-CD86-V450 (clone GL-1), anti-CD4-FITC (clone RPA-T4), and anti-CD8-PE (clone RPA-T8, BD Biosciences, San Jose, CA, USA). All the samples were treated with propidium iodide to gate dead cells. Flow cytometry was performed with a FACSCalibur (BD Biosciences, San Jose, CA, USA). Cytokines in mice sera, DC, or MoDC supernatants were quantified using multiparametric Luminex kits. In brief, IFN-gamma, IL-2, IL-4, IL-6, IL-10, IL-12 (p70), IL-17A, KC/CXCL1, MIP-2, and TNF-α levels in mice serum samples were quantified using the Luminex 200 platform with a magnetic system (Milliplex MAP Mouse High Sensitivity T Cell Magnetic Bead Panel, EMD Millipore Corporation, Billerica, MA, USA) following the manufacturer’s instructions. Cytokine concentrations are expressed as the averages of three replicates in pg/mL ± SD. Human cytokines in MoDC supernatants were quantified using the multiparametric Luminex kit (Milliplex human HSTCMAG-28SK including the following cytokines: IFN-γ, IL-10, IL-17A, IL-2, IL-4, IL-6, and TNF-α; EMD Millipore Corporation, Billerica, MA, USA) following the manufacturer’s instructions.

### 2.8. Statistical Analysis

For statistical analysis, Student’s *t*-test was applied to mice with auto transplants; each group included 5 mice for all assays reported (*p* ≤ 0.05 was considered significant). Experiments in mice were performed at least five times each, and experiments with MoDCs from healthy donors or oncologic patients and tumor cell lines were performed three times each. ANOVA analysis was applied to cytokine measurements and flow cytometry analysis per the manufacturer´s recommendations (*p* ≤ 0.05 was considered significant). For statistical purposes, each flow cytometry sample was performed in triplicate. The GraphPad software was used for the generation of all the graphs presented.

## 3. Results and Discussion

We initiated this study with the hypothesis that GNP-LLO_91–99_ nanovaccines might function as immunotherapies for bladder tumors and planned the following assays: adjuvant abilities using MoDCs from human donors, the in vitro immunogenic death of murine and human tumor cell lines, the in vivo anti-neoplastic action after the establishment of a mouse bladder cancer model, and immunotherapeutic actions in comparison to anti-CTLA-4 or anti-PD-1 immunotherapies.

### 3.1. GNP-LLO_91–99_ Nanovaccines Showed No Toxicity in Human MoDCs or Mice

Before initiating any experiment, we prepared a homogeneous batch of GNP-LLO_91–99_ nanovaccines to be used in all the experiments and performed the quality and toxicity controls in vivo and in vitro. GNP-LLO_91–99_ nanovaccines are formed by a gold core covalently linked to LLO_91–99_ peptide and β-D glucose (GNP-LLO_91–99_ nanovaccines schematic representation in Figure 1a). The gold core had a spheric shape and nanometric size with an average of 2 nm, as observed by the analysis of transmission electron microscopy (Figure 1b). Different concentrations of GNP-LLO_91–99_ nanovaccines or LLO_91–99_ peptide (5–500 µM) were non-toxic for human MoDCs after incubation for 16 h, as determined by evaluating the cell viability with Trypan blue staining (Figure 1c). We also observed no toxicity in mice (C52BL/6 male and female, n = 10) when we inoculated intraperitoneally (i.p.) for 16 h with GNP-LLO_91–99_ nanovaccines or LLO_91–99_ peptide (5–500 µM) (Figure 1c, center and right columns) and examined mice health conditions and sera IL-1 concentrations.

### 3.2. GNP-LLO_91–99_ Nanovaccines Served as Adjuvants for Human MoDCs of Oncologic Patients

Once we established that GNP-LLO_91–99_ nanovaccines showed no toxicity in human MoDC, we next explored their abilities as adjuvants. Functional and activated MoDCs are characterized by three molecules relevant to combat tumors, the surface expression of MHC-I and MHC-II molecules necessary for antigen presentation, and CD86 co-stimulatory molecules essential for T cell activation. MoDC cytokine production is also a determinant of T cell activation or suppression. In this regard, MoDC activation corresponds with a high production of Th1 cytokines, such as IL-12p70, which stimulates cytotoxic T cells, or TNF-a, which implements innate immunity. Meanwhile, suppression is induced by classical Th2 cytokines, such as IL-6 or IL-10, which promote significant numbers of T_reg_ [19] as well as KC or MIP-2 chemokines that also participate, as they can trigger the migration of MDSC. The ability of an adjuvant to induce both maturation and activation of ex vivo cultured DCs requires an optimal and basal functionality of these cells [30], which is not always observed in cancer patients, as reported in hepatocarcinoma or multiple myeloma [12,13,31,32]. For this reason, we evaluated GNP-LLO_91–99_ nanovaccines as adjuvants after the ex vivo incubation of MoDCs from healthy donors or patients with melanoma (MEL-1, MEL-2) or bladder carcinoma (BC-1, BC-2) from our institution (Hospital U. Marqués de Valdecilla, HUMV, Santander, Spain) (Appendix A). GNP-LLO_91–99_ nanovaccines showed three-fold increases in the percentages of MHC-I or MHC-II and five-fold increases in the co-stimulatory CD86 molecules after incubation of MoDCs in healthy donors (CONT), patients with melanoma (MEL-1, MEL-2), lung and bladder tumors (BC-1), or urothelial bladder tumors (BC-2) (GNP-LLO_91–99_ bars in Figure 2a). Empty GNP nanovaccines or LLO_91–99_ peptides (data not shown) had no effect on the surface expression of MHC-I, MHC-II, or CD86 molecules, similar to MoDCs treated with saline (CONT + GNP and CONT bars in Figure 2a), as reported previously [7].

Analysis of Th1 and Th2 cytokines indicated that GNP-LLO_91–99_ nanovaccines increased the production of Th1 cytokines in MoDCs of healthy donors, especially TFN-α and IL-12p70 (CONT versus CONT + GNP-LLO_91–99_ rows in Table 1). Empty GNP nanovaccines showed similar patterns as saline-treated MoDCs (CONT versus CONT + GNP rows in Table 1), indicating that empty GNP showed no adjuvant effect.

MoDCs from patients with melanoma or lung and bladder carcinoma showed under non-stimulated conditions a clear Th2 cytokine pattern (e.g., IL-6 and IL-10, high levels) that reflects the systemic cytokine pattern of the patient’s sera (Appendix A). Incubation of MoDCs with GNP-LLO_91–99_ nanovaccines shifted Th2 to Th1. In fact, we detected high levels of IL-12p70 and reduced levels of IL-10 (MEL-1-GNP-LLO_91–99_, MEL-2-GNP-LLO_91–99_, BC-1-GNP-LLO_91–99_, and BC-2-GNP-LLO_91–99_ rows in Table 1). Therefore, GNP-LLO_91–99_ nanovaccines seem to be valid adjuvants for patients with melanoma or BC but also for MoDCs of patients with other tumors (e.g., prostate adenocarcinoma, NSCLC lung cancer, or glioblastoma), except for those reported with a low DC functionality such as hepatocellular carcinoma [13] (Appendix A).

### 3.3. GNP-LLO_91–99_ Nanovaccines Induced Immunogenic Apoptosis in Bladder Tumors

Once we evaluated the possibility that GNP-LLO_91–99_ nanovaccines served as efficient adjuvants, we examined their induction of tumor apoptosis. For this purpose, we used available tumor cell lines of melanoma (murine B16.F10 or human A-375 cell lines) or BC (murine MB-49 or human T-24 cell lines). We examined two types of apoptosis by FACS: direct apoptosis and immunogenic DC-mediated apoptosis. GNP-LLO_91–99_ nanovaccines induced low percentages of direct apoptosis onto BC or melanoma tumor cell lines, barely 5% in murine B16.F10 melanoma (black bars in Figure 2b), and showed no cytotoxicity (Appendix A). Assays of DC-mediated immunogenic apoptosis imply incubation of tumor cell lines with ½ of the supernatants of DCs or MoDCs, pre-incubated with GNP-LLO_91–99_ nanovaccines. GNP-LLO_91–99_ nanovaccines induced high percentages (35–55%) of immunogenic apoptosis in melanoma and BC (gray bars in Figure 2b). GNP-LLO_91–99_ induction of immunogenic apoptosis seemed effective for tumors with significant T cell infiltration such as melanoma, BC, NSCLC lung tumors (murine TC-21 or human A-549 cell lines), or glioblastoma (murine O627 or human RG-1 cell lines), while not in tumors with low T cell infiltration, such as in hepatocellular carcinoma (murine Hepa 1-6 cell lines) [12,13], ovary tumors (hamster CHO cell lines), or SV-40-induced tumors (murine IC-21 macrophage-like cell lines) (Appendix A). We conclude that GNP-LLO_91–99_ nanovaccine induction of DC-mediated immunogenic apoptosis appears to explain their anti-neoplastic abilities.

### 3.4. Mechanisms of Anti-Neoplastic Abilities of GNP-LLO_91–99_ Nanovaccines in Mice Models of Melanoma and BC

To validate the anti-neoplastic abilities of GNP-LLO_91–99_ nanovaccines for BC, we established subcutaneous (s.c.) transplants of murine bladder MB-49 and melanoma B16.F10 tumor cell lines (protocol for all the following experiments is shown in Figure 3a) for 14 days, showing a size of 400–500 m^3^.

First, we performed experiments to examine the effects of a single dose of GNP-LLO_91–99_ nanovaccines at different times, i.e., 7, 14, and 23 days post-treatment, and observed similar results (Table 2, files labeled as + GNP-LLO_91–99_). Mice were intravenously (i.v.) inoculated in their tails with a single dose of GNP-LLO_91–99_ nanovaccines (50 µg/mouse), and 7 days post-treatment, mice were sacrificed, and sera and tumors were collected to examine the effects of nanovaccines (Figure 3b). GNP-LLO_91–99_ nanovaccines reduced 5-fold tumor volume of B16.F10 melanoma and 4.7-fold of MB-49 bladder cell lines (GNP-LLO_91–99_ bars in Figure 3b). The sizes of control tumors were 10-fold larger than tumors at 7 days, and survival of mice was 60% and 80% reduced, respectively, presenting tumor ulcerations (Table 2, SR and U columns). To avoid unnecessary pain to mice, for further experiments, we established a protocol of 14 days transplantation and 7 days treatment with GNP-LLO_91–99_ nanovaccines (Figure 3a). GNP-LLO_91–99_ nanovaccines implemented the survival rates (Table 2, SR columns of files labeled as + GNP-LLO_91–99_) and blocked tumor growth at all times tested (TV columns).

Second, we explored the immune mechanisms able to explain the anti-neoplastic abilities of GNP-LLO_91–99_ nanovaccines in BC mouse models, analyzing the cytokines in mouse sera. We confirmed a Th2 cytokine versus a Th1 pattern in mice transplanted with bladder MB-49 cell lines (light gray bars in Figure 4a), similar to the pattern detected with melanoma B16.F10 [6,7] (white bars in Figure 4a). In this regard, we detected high levels of chemokines recruiting neutrophils such as KC/CXCL1, participating in MDSC and tumor progression such as MIP-2, and high levels of cytokines such as IL-6 and TNF-a cytokines, while low levels of pro-inflammatory Th1-Th17 cytokines such as IFN-γ, IL-2, and IL-17A (CONT-B16F10 and CONT-MB-49, white and light gray bars in Figure 4a). Therefore, systemic production of chemokines/cytokines in mice transplanted with bladder MB-49 cell lines suggested an immunosuppression status [33] similar to the levels of BC or melanoma patients (Appendix A).

However, innate immune responses were not identical in melanoma and BC, as we detected a high production of IL-17A cytokines in melanoma and normal levels in BC mice (CONT-B16F10 white bars versus CONT-MB-49 light gray bars Figure 4a). The treatment of transplanted mice with GNP-LLO_91–99_ nanovaccines decreased the levels of Th2 cytokines KC, MIP-2, IL-6, and TNF-α and increased the levels of IFN-γ and IL-2, confirming the shift of Th2 to Th1 cytokine pattern in both mouse models of tumor transplantation, B16.F10 melanoma and MB49 bladder cell lines, suggesting a common action (black and dark gray bars in Figure 4a,b).

### 3.5. GNP-LLO_91–99_ Nanovaccines Are Effective Immunotherapies That Blocked Immunosuppression

Next, we confirmed the immunosuppression status after isolation of tumor-infiltrated lymphocytes (TILs) of mice s.c. transplanted with melanoma or BC, and treated or not with GNP-LLO_91–99_ nanovaccines. As shown in Figure 5a, BC basal status showed a higher immunosuppressive environment than melanoma, with higher levels of T_reg_ and MDSC (Ly6G^+^CD11b^+^) and lower levels of activated DCs (CD11c^+^MHC-II^+^) and macrophages (F4/80^+^) (compare light gray versus white bars in Figure 4a).

GNP-LLO_91–99_ nanovaccine action reduced immunosuppressed cells (T_reg_, MDSC) in both tumor models and increased activated DCs and macrophages (compare black and dark gray bars in Figure 5a). Therefore, GNP-LLO_91–99_ nanovaccines eliminated immune blockers in BC and melanoma, suggesting that they might function as immunotherapies for BC, as reported for melanoma [7].

Next, we verified that GNP-LLO_91–99_ nanovaccines acted as immunotherapies in mice models of BC, exploring the immune populations in TILs of mice inoculated with a single dose of GNP-LLO_91–99_ nanovaccines alone (50 µg/mouse) or in combination with anti-CTLA-4 (50 µg/mouse) or anti-PD-1 (50 µg/mouse) immunotherapies. GNP-LLO_91–99_ nanovaccines, alone or in combination with anti-CTLA-4 or anti-PD-1 antibodies, increased the percentages of DCs with an activated MHC-II^+^CD40^+^CD86^+^ phenotype of functional APC and the numbers of T and B cells (e.g., CD4^+^, CD8^+,^ and CD19^+^ cells), while these treatments reduced the percentages of suppressor innate cells such as MDSC (Ly6G^+^CD11b^+^ granulocytes) and T_reg_ (CD25^+^ cells) (compare black, light gray, and dark gray bars in Figure 5b). We also included as controls mice treated with anti-CTLA-4 or anti-PD-1. As shown in Figure 5b, ICI alone cannot induce the percentages of activated DCs observed in GNP-LLO_91–99_ nanovaccines combined with ICI. Anti-CTLA-4 treatment alone caused a moderate reduction in T_reg_ but not as prominent as observed with GNP-LLO_91–99_ nanovaccines combined with anti-CTLA-4.

In summary, GNP-LLO_91–99_ nanovaccines appear to be valid immunotherapy for BC as well as melanoma. Moreover, the ability of GNP-LLO_91–99_ nanovaccines to induce immunogenic apoptosis in solid tumors with high or low T cell infiltration, such as NSCLC lung cells or glioblastoma, and to activate MoDCs from patients with T cell-infiltrated solid tumors, such as prostate adenocarcinoma, multiform glioblastoma, or lung carcinoma, opens the possibility for future validation of GNP-LLO_91–99_ nanovaccines as a neoadjuvant or immunotherapy for tumors with high or medium T cell infiltration.

## 4. Conclusions

GNP-LLO_91–99_ nanovaccines appear to act as adjuvants for BC patients, as they promoted (i) the antigen presentation capacities of MoDCs of patients with BC (e.g., increasing the levels of MHC-I and MHC-II molecules as well as co-stimulatory CD86 molecules) and (ii) a Th1 cytokine pattern by releasing high levels of IL-12p70 and TNF-α with anti-neoplastic abilities.

In this study, GNP-LLO_91–99_ nanovaccines presented anti-neoplastic abilities for BC, as they induced immunogenic apoptosis directed by dendritic cells and the cytokines released by them.

Moreover, GNP-LLO_91–99_ nanovaccines showed immunotherapeutic abilities for BC, as they blocked the immunosuppression status of BC, increasing the numbers of cytotoxic T cells and DCs within the tumors and decreasing the number of immunosuppressive cells (T_reg_, MDSC). In fact, GNP-LLO_91–99_ nanovaccines adequately combine well and potentiate the action of ICI—both anti-CTLA-4 and anti-PD-1—supporting them as a novel nano-immunotherapy for BC.

## 5. Patents

F.R., S.Y.D. and C.A.D. participated as authors in the patent number WO2017144762 https://patentscope.wipo.int/search/es/detail.jsf?docId=WO2017144762 (accessed on 9 May 2022) and US20190045527 https://patentscope.wipo.int/search/es/detail.jsf?docId=US237388381&_cid=P21-KC8ZLF-10597-1 (accessed on 9 May 2022), entitled to Instituto de Investigación Marqués de Valdecilla (IDIVAL).

## Figures and Tables

**Figure 1 cancers-14-02413-f001:**
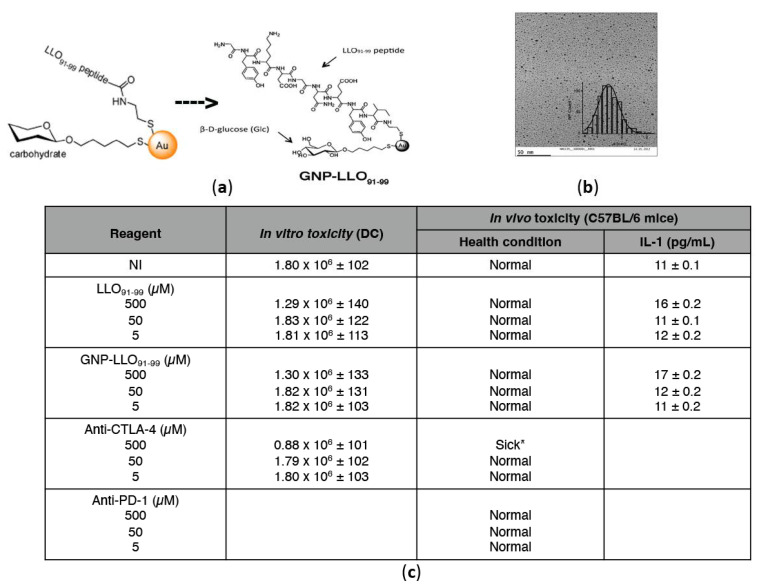
Preparation of GNP-LLO_91–99_ nanovaccines, quality, and toxicity controls in vitro and in vivo. (**a**) Schematic representations of GNP-LLO_91–99_ nanovaccines and chemical structure showing localization of LLO_91–99_ peptide and the carbohydrate, β-D-glucose. (**b**) Transmission electron microscopy (TEM) image (100,000×), including the size histogram that shows the spherical shape and nanometric size of the gold core of GNP-LLO_91–99_. (**c**) In vitro toxicities of GNP-LLO_91–99_ nanovaccines were explored in MoDCs of healthy donors incubated for 16 h with different concentrations of GNP-LLO_91–99_ nanovaccines or LLO peptide, LLO_91–99_ (5–500 µM) to examine cell viability with Trypan-blue staining. Results are expressed as the mean of viable cells ± SD. For in vivo analyses, we inoculated mice (12 months old male, n = 5, or female, n = 5) intravascularly (i.v.) with the same concentrations of GNP-LLO_91–99_ nanovaccines and LLO_91–99_ peptide as in (**a**) (5–500 µM) in C57BL/6 mice for 16 h, examined mice health conditions, and measured IL-1 concentration in mice sera. Experiments were performed five times each. Results are expressed as the mean cytokine concentrations (pg/mL) ± SD. * Sick, refers to animals with hairless and difficulties to move and feed.

**Figure 2 cancers-14-02413-f002:**
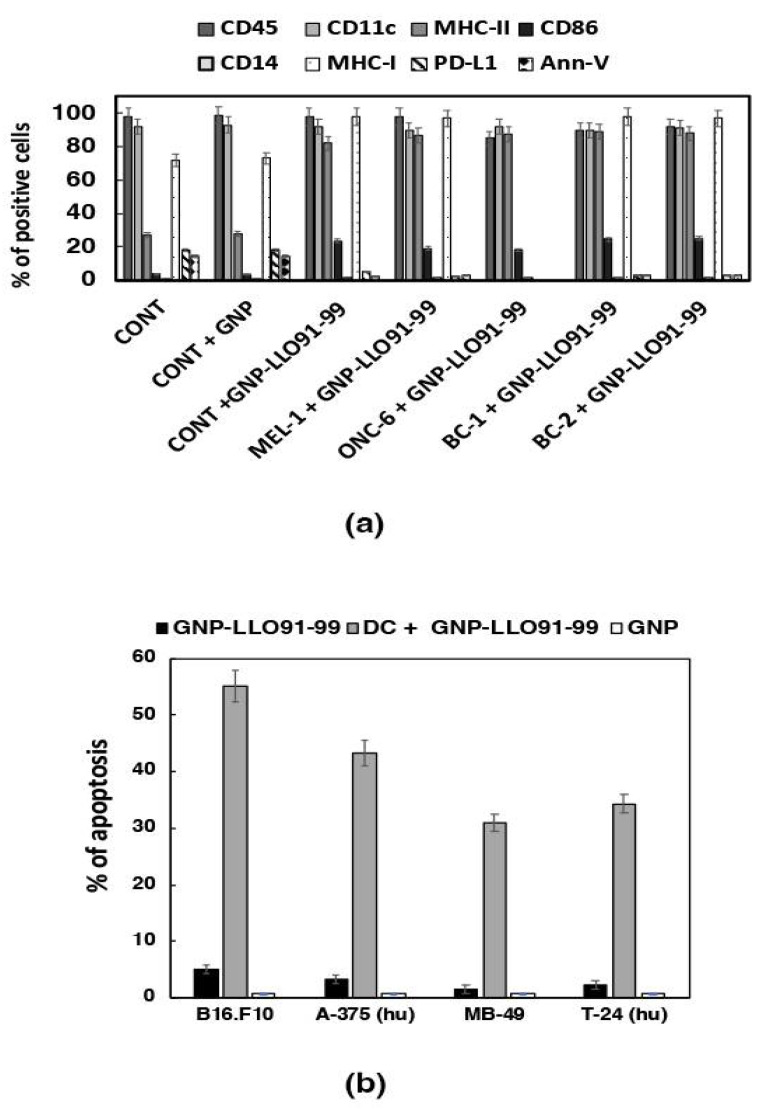
Adjuvant effects and immunogenic apoptosis abilities of GNP-LLO_91–99_ nanovaccines. (**a**) Ex vivo differentiated MoDCs from patients (melanoma: MEL-1, MEL-2, bladder cancer: BC-1, BC-2) or healthy donors (CONT) with the phenotype CD45^+^CD11c^+^CD14^-^ were incubated or not with GNP-LLO_91–99_ nanovaccines (+GNP-LLO_91–99_ bars). FACS was performed to analyze cell surface markers, and results are expressed as the mean percentages of positive cells ± SD (*p* ≤ 0.05). (**b**) Murine B16.F10 and human A-375 melanoma, and murine MB-49 and human T-24 bladder tumors were incubated with GNP-LLO_91–99_ for direct apoptosis (black bars) or ½ supernatants of DCs/MoDCs pre-treated with GNP-LLO_91–99_ nanovaccines for immunogenic apoptosis (gray bars). Apoptosis was examined by FACS using the DNA marker, 7-AAD (7-AAD-PE), and the apoptotic marker, annexin V (annexin V-APC). Experiments were performed at least three times. Results are expressed as percentages of apoptotic cells ± SD (*p* ≤ 0.05).

**Figure 3 cancers-14-02413-f003:**
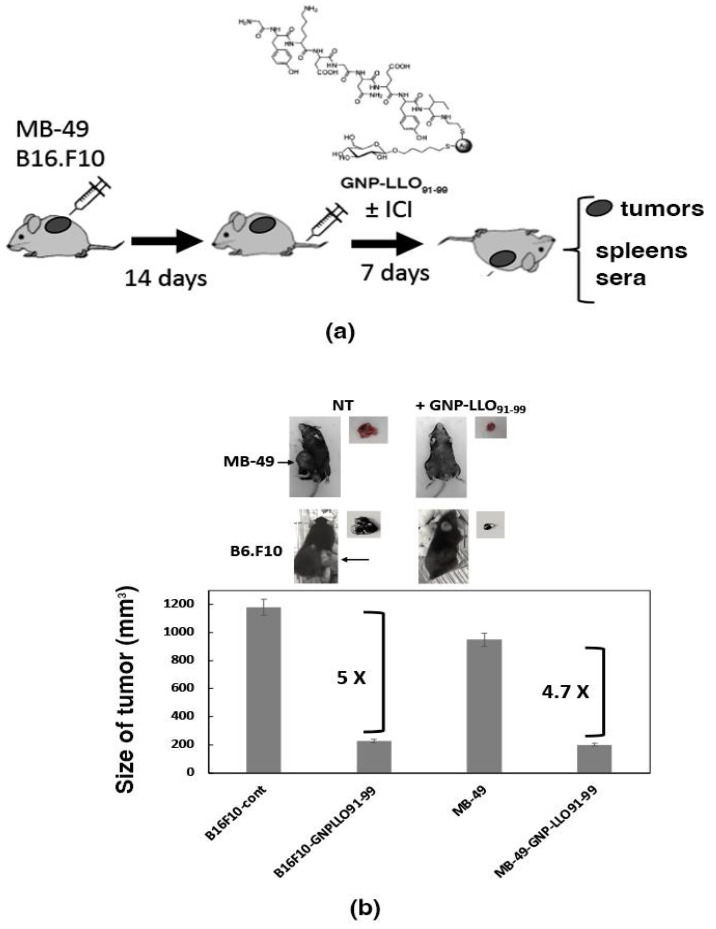
Anti-neoplastic abilities of GNP-LLO_91–99_ nanovaccines. (**a**) General scheme of GNP-LLO_91–99_ nanotherapy for all types of experiments. Melanoma (B16.F10) or BC (MB-49) tumor cell lines were s.c. transplanted into the right hind flanks of mice for 14 days. Next, mice were inoculated i.v. with a single dose of GNP-LLO_91–99_ (50 µg/mouse) in the presence or absence of anti-CTLA-4 or anti-PD-1 antibodies (here labeled as ICI), or left untreated (CONT bars), and were sacrificed 7 days later. (**b**) GNP-LLO_91–99_ treatment caused tumor reduction in B16.F10 melanoma and MB49 bladder tumor cell lines (upper images show the tumors isolated from mice, and lower graphs show the sizes measurements of isolated tumors).

**Figure 4 cancers-14-02413-f004:**
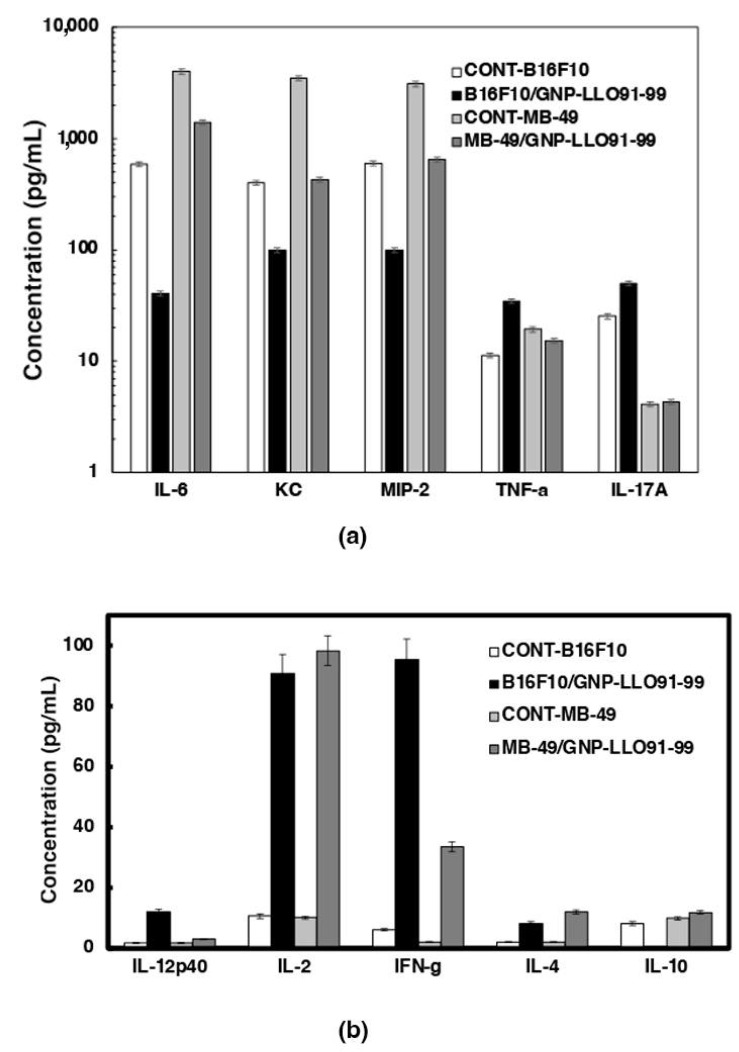
GNP-LLO_91–99_ nanovaccines activate Th1 responses in mice models of melanoma and BC. Cytokines were measured in sera of mice transplanted with melanoma or BC and treated with GNP-LLO_91–99_ nanovaccines (melanoma: white, BC: light gray bars) or untreated (melanoma: black, BC: dark gray bars) using a multiparametric Luminex kit (Milliplex MAP Mouse High Sensitivity T Cell Magnetic Bead Panel, EMD Millipore Corporation, Billerica MA) following the manufacturer’s instructions. Cytokine concentrations are expressed as the average of three replicates in pg/mL ± SD. (**a**) Cytokines and chemokines produced by innate immune cells. (**b**) Cytokines produced by lymphocytes. Experiments were performed five times.

**Figure 5 cancers-14-02413-f005:**
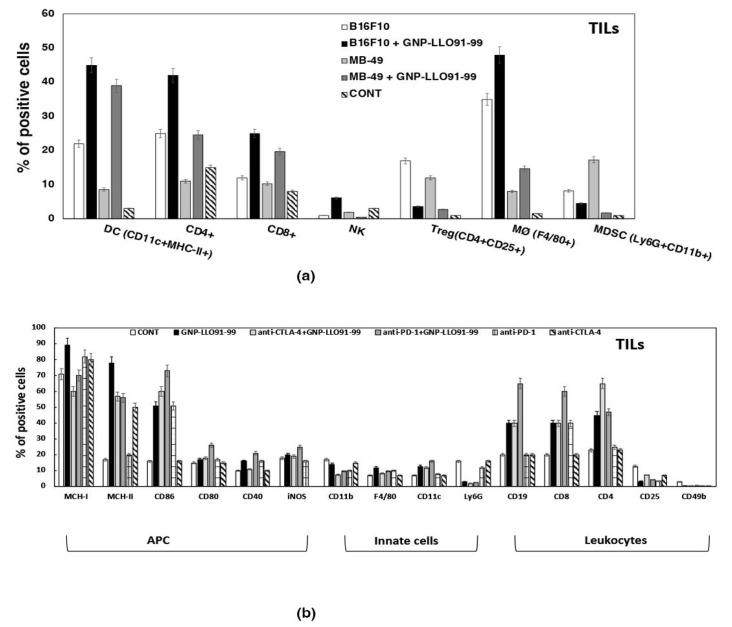
GNP-LLO_91–99_ nanovaccines are effective immunotherapies in mice models of BC. (**a**) Mice s.c. transplanted with B16.F10 melanoma or MB-49 BC cell lines and treated or not with GNP-LLO_91–99_ nanovaccines as in Figure 3 were sacrificed, tumors were collected, and TILs were isolated from interphases of Ficoll gradients (untreated mice: white bars for melanoma and light gray bars for BC; GNP-LLO_91–99_-treated mice: black bars for melanoma and dark gray bars for BC). FACS was performed to analyze cell surface markers and results are expressed as the mean percentages of positive cells ± SD (*p* ≤ 0.05). (**b**) Mice s.c. transplanted with MB-49 bladder tumor cell lines and treated or not (CONT bars) with GNP-LLO_91–99_, alone (black bars) or the combination of anti-CTLA-4 antibodies (light gray bars) or anti-PD-1 antibodies (dark gray bars). TILs were collected, and cell surface markers were analyzed by FACS. Results are expressed as the mean percentages of positive cells ± SD (*p* ≤ 0.05).

**Table 1 cancers-14-02413-t001:** Production of cytokines by MoDCs stimulated or not with GNP-LLO_91–99_ nanovaccines.

Patient Code	Clinical Symptoms and Treatment ^a^	IL-12 ^b,^*	IL-6	IL-10	TNF-α
MEL-1	Metastatic melanoma (IV) NT	0.3 ± 0.1	6.4 ± 0.9	5.6± 0.9	6.25 ± 0.1
MEL-1 + GNP-LLO_91__–__99_	Metastatic melanoma (IV) NT	4.7 ± 0.1	2.22 ± 0.2	1.61 ± 0.1	66 ± 1.3
MEL-2	Nodular melanoma (IIIB) surgery	0.5 ± 0.1	6.1 ± 0.1	6.6 ± 0.1	8.78 ± 0.1
MEL-2 + GNP-LLO_91__–__99_	Nodular melanoma (IIIB) surgery	6.5 ± 0.2	2.07 ± 0.1	1.7 ± 0.1	50 ± 1.2
BC-1	Lung–bladder carcinoma Cisplatin–etoposide	0.2 ± 0.1	6.0 ± 0.1	4.6 ± 0.1	7.8 ± 0.1
BC-1 + GNP-LLO_91__–__99_	Lung–bladder carcinoma Cisplatin–etoposide	6.4 ± 0.1	2.3 ± 0.1	1.2 ± 0.1	220 ± 1.4
BC-2	Urothelial bladder carcinoma NT	0.3 ± 0.2	7.1 ± 0.2	5.1 ± 0.2	8.1 ± 0.1
BC-2 + GNP-LLO_91__–__99_	Urothelial bladder carcinoma NT	7.2 ± 0.3	2.1 ± 0.1	0.9 ± 0.2	204 ± 1.6
CONT	NONE	0.8 ± 0.1	3.1 ± 0.1	2.4 ± 0.1	2.0 ± 0.1
CONT + GNP	NONE	0.7 ± 0.1	2.8 ± 0.1	2.3 ± 0.1	1.8 ± 0.1
CONT + GNP-LLO_91__–__99_	NONE	6.0 ± 0.2	4.1 ± 0.2	3.5 ± 0.1	150 ± 0.2

^a^ Clinical symptoms and treatments of patients with informed consent selected for the study. CONT, healthy donors. ^b^ Cytokines are measured in sera of patients (Luminex kits, EMD Millipore Corporation, Billerica, MA, USA). Results are the mean of cytokine concentrations (pg/mL) ± SD (* *p* ≤ 0.05).

**Table 2 cancers-14-02413-t002:** Clinical parameters of mice transplanted with melanoma and bladder tumors after GNP-LLO_91–99_ therapy.

Tumor Cell Line and Treatment ^a^	Day 7			Day 14			Day 23		
	SR ^c^	U	TV ^d^	SR	U	TV	SR	U	TV
MB49 + GNP-LLO_91–99_ ^b^	100% ± 1	-	426 ± 5	100% ± 1	-	436 ± 11	100% ± 1	-	490 ± 10
B16.F10 + GNP-LLO_91–99_	100% ± 1	-	460 ± 2	100% ± 2	-	470 ± 13	100% ± 2	-	490 ± 9
MB49-NT	85% ± 2	-	1080 ± 5	45% ± 5	+/−	1680 ± 16	16% ± 2	+	4921 ± 19
B16.F10-NT	84% ± 2	-	1090 ± 6	42% ± 2	+/−	1690 ± 19	10% ± 2	+	5322 ± 20

^a^ C57BL/6 female mice were s.c. transplanted with B16.F10 melanoma cells or MB49 bladder tumor cells (n = 10 mice/group) for 14 days showing a size of 400 m^3^ for B16.F10 melanoma and 460 m^3^ for MB49 bladder tumors. ^b^ Tumor transplanted mice received a single dose of GNP-LLO_91–99_ i.v. (50 µg/mice) or saline (non-treated, NT) for 7 days. Number of surviving mice was counted at days 7, 14, or 23 post-transplantation. ^c^ Survival rates (SR) are expressed as the mean ± SD, and ulcerations (U) are indicated as positive (+) for abundant and severe ulcerations, (+/−) for rare and mild ulcerations, or negative (-) for absence of ulcerations. ^d^ Mice treated as in b were sacrificed, and tumors were recovered and measured with a caliper. Tumor volumes (TV) were calculated with the following formula: (length × (width)^2^)/2. Results are the mean ± SD of tumor volumes. All experiments were repeated five times. Results are the mean ± SD (*p* ≤ 0.05).

## Data Availability

The data presented in this study are available on request from the corresponding author.

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
