# Peer review of "Gold Glyconanoparticles Combined with 91–99 Peptide of the Bacterial Toxin, Listeriolysin O, Are Efficient Immunotherapies in Experimental Bladder Tumors"

_cancers, 2022, doi:10.3390/cancers14102413_

Round 1

Reviewer 1 Report

The authors describe results of bladder tumor therapies by gold glyconanoparticles combined to peptide. Mice as well as patients with tumors were treated with the nanovaccines.

The manuscript is a continuation of work of the group. It is well designed and the results are presented seriously.    

In the Materials section preparation of gold nanoparticles and nanovaccine is missing.

Author Response

We thank the reviewer suggestion regarding that in the Materials section, the preparation of gold nanoparticles and nanovaccines was missing and moved the data and Figure of the Supplemental file (Figure S1) to the main text, now Figure 1 and included the new section in Materials and Methods as follows:

---2.3. Preparation of GNP-LLO91-99 nanovaccines. LLO91–99 peptide with a C-terminal cysteamide (LLO91–99C(O)NHCH2CH2SH, 14 mg, purity 95%) was purchased from GenScript (Piscataway, NJ) and 5-(mercapto)pentyl-β-D-glucopyranoside (GlcC5SH) was prepared as reported previously by our group [27]. GNPs carrying 90% glucose and 10% LLO91–99 peptide are prepared by the reduction of Au(III) salt with sodium borohydride following previously described procedures [28]. Peptide loading was GNP-LLO91-99: 8.9 µg (LLO91–99)/0.182 mg of GNPs. In brief, LLO91–99C(O)NHCH2CH2SH (1 mg, 0.85 µmol) and GlcC5SH (2.1 mg, 7.4 µmol) were dissolved in deuterated water (750 µL). 1H NMR analysis of the mixture showed a Glc:LLO91–99 ~ 9:1 ratio (e.g. 90% of Glc respect to 10% of LLO91–99). The above-prepared solution of the ligands (0.011 M, 4 equivalents) was added with an aqueous solution of HAuCl4 (100 μL, 0.025 M, 1 equivalent) followed by an aqueous NaBH4 solution (67.5 μL, 1 M, 27 equivalents) in four portions under rapid shaking. The dark dispersion was shaken for 2 h, and then filtered with 3 KDa MWCO tubes by centrifugal filtering. The black colloid was recovered with water and lyophilized (0.482 mg). The residue was re-dispersed in the minimum volume of water, loaded in a SnakeSkin pleated dialysis tubing (Pierce, 10KDa MWCO) and dialyzed against 3 L of water under gentle stirring, recharging with fresh distilled water every ~8 h over the course of 72 h. After lyophilization, 0.456 mg of GNP-LLO91-99 were obtained. The ratio of glucose/peptide in the GNP was determined by quantitative NMR (qNMR) in a Bruker AVANCE 500 MHz spectrometer: 0.456 mg of GNP-LLO91-99 were dispersed in D2O 99.9% (200 µL). 80 µL of this solution were added with 40 µL of a 0.05% 3-(trimethylsilyl)propionic-2,2,3,3-d4 acid sodium salt (TSP-d4) solution in D2O and 60 µL of D2O. 1H NMR analysis of the mixture allowed the calculation of the amount of peptide in the GNP-LLO91-99: 8.9 µg (LLO91–99)/0.182 mg (GNP). TEM (JEOL JEM-2100F working at 200 kV): A single drop (5 μL) of the aqueous dispersion (ca. 0.1 mg mL−1 in MilliQ water) of the GNPs was placed onto a copper grid coated with an ultrathin carbon film (Electron Microscopy Sciences). The grid was left air-dried for 12 h at RT. Average gold diameter 1.5 ± 0.5 nm, obtained by counting 600 particles in the TEM image as previously reported [28]. UV/Vis (Beckman Coulter DU 800 Spectrometer, H2O, 0.1 mg/mL): The average gold core size was confirmed by UV-Vis spectra, which did not show an absorption maximum at around 520 nm, typical of gold nanoparticles with a bigger core diameter [29].

---New Figure 1 in the revised manuscript presents all the quality tests performed with the nanoparticles, and their lack of toxicities in vitro using monocyte derived dendritic cells of healthy donors and in vivo assays in mice and reflected no toxicity, as determined by the production of an acute cytokine as IL-1 and exploring clinical parameters of healthy mice such as normal motility and feeding, no lack of hair and bright hair appearance.

We hope to have satisfied the reviewer's concern

Reviewer 2 Report

The manuscript by Terán-Navarro et al. seeks to present evidence of gold nanoparticles loaded with the bacterial peptide 91-99 of the listeriolysin O toxin being a candidate immune therapeutic against cancer. While data in the manuscript seem so support this notion to a certain degree, the experimental design suffers from two major flaws. First, there no indication that key in vivo experiments (these using mouse models) were repeated more than once and therefore no conclusions on data reproducibility can be drawn.As a side note, using a single time-point (day 7 post treatment, see Fig. 2) readout is also not optimal if not misleading. Second (and most important), key control groups like nanoparticles w/o peptide, peptide alone, immune checkpoint inhibitors (ICI) alone are completely missing and therefore it is completely impossible to figure out how functional proposed therapeutic really is, i.e. is it synergistic with ICI or antagonistic (Fig. 4b)? Would MoDC be stimulated by gold particles w/o listeriolysin peptide (Fig. 1a)? Such questions pertain to nearly all datasets presented, not to mention that Fig. 2 shows that ICI was indeed used for  treatment of tumor-bearing mice in combination with listeriolysin peptide-carrying nanoparticles, but there's no indication what was its effect, if any.

There are also a number of language and terminological inaccuracies such as repeated use of word 'immunotherapies' instead of correct singular 'immunotherapy', 'is the nine most common malignancy' (line 77) instead of '9th most common' or 'implements' on line 201, typos (enrollment with 'l' missing on line 107) and pure sloppiness ('p<0.5' on line 183 instead of p<0.05). As a further example of all of it coming together is the sentence in lines 210-216, which is completely unreadable and uses the expression 'shows increases in the percentages' instead of simply 'increases'. However, al of these defects can be fixed, but the flaws of experimental design and execution cannot.

Collectively, said nanoparticles containing listeriolysin O peptide may be indeed immunostimulatory, but to what extent they may be useful in real world cancer therapy is a major question, which is currently left unanswered.

Author Response

We appreciate the reviewer comments and have performed the following changes to improve the quality of the manuscript:

  • We have extended the Introduction section to include the explain the effect of listeriolysin O to induce cytolysis and cell death and the type of cell death as an important effect we have followed with the listeriolysin O derived GNP nanoparticles for bladder tumor and included 3 new references (new references 24, 25 and 26).
  • We have indicated the times each experiment has been repeated in each figure to indicate that our conclusions are sustained by the reproducibility of our data. In general, mice experiments have been performed at least five times. While data with blood cells of patients or sera have been repeated three times, since human samples are limited and we were not allowed to ask for more samples from a single patient according to the Informed Consents.
  • We have moved Figure S1 in the Supplemental file to the main text (now Figure 1) to present all the quality tests performed with the nanoparticles, and their lack of toxicities in vitro using monocytes derived dendritic cells of healthy donors and in vivo assays in mice and reflected no toxicity as determined by the production of an acute cytokine as IL-1 and exploring clinical parameters of healthy mice such as normal motility and feeding, no lack of hair and bright hair appearance.
  • We have included as controls of the results obtained with GNP-LLO91-99 nanovaccines, empty GNP in the following new remodeled figures: new Figure 2 (panels a and b) and new Table 1 and detected that empty GNP showed no adjuvant effect in monocyte derived dendritic cells (MoDC) (panel a of new figure 2) and no induction of immunogenic apoptosis, either (panel b of new figure 2). LLO91-99 peptide alone had no effect either as described in reference 7.
  • We have explained that the protocol shown in new Figure 3 (panel a) is a general protocol for all the different experiments using GNP-LLO91-99 nanovaccines in mice models of tumors.
  • We have included a new Table, new Table 2, to include different time-points for the treatment GNP-LLO91-99 nanovaccines in mice transplanted with melanoma or bladder tumor cells and explained why we have selected as readout 14 days post-transplantation since tumors at longer times (23 or 30 days) showed uncontrolled growth and cause too much pain to mice. GNP-LLO91-99 treatment for 7 days was sufficient to see the anti-neoplastic effect.
  • We have included single treatments with each of the ICI used, anti-CTLA-4 and anti-PD-1 in the experiments performed of GNP-LLO91-99 nanovaccines combination with ICI (new Figure 5, panel b). None of the ICI alone caused an increased in the percentages of activated DC within the TILs of tumors. As it was previously reported for melanoma (reference 7), we also observed that treatment with anti-CTLA-4 alone, caused a moderate reduction of Treg but not as prominent as the combination of GNP-LLO91-99 nanovaccines and anti-CTLA-4.
  • We have corrected all the misspellings and terminological inaccuracies: as immunotherapies instead of immunotherapy or the error typos (p≤0.5 instead of p≤0.05) and remodeled some sentences for simplicity.
  • We have included British English as the language for the DOC revision and now there is no misspellings or bad English style. We have also given the manuscript to an English speaker. If reviewer consider is not sufficient, we will submit the text to an English Editing Company.

We hope these changes and comments can satisfy the reviewer concerns regarding our manuscript.

Round 2

Reviewer 2 Report

Authors have extensively revised and improved their manuscript. Indicating how many times a particular experiment was repeated is especially helpful. The manuscript can now be accepted for publication in the present form.